# Prevalence and Characteristics of Sexual Victimization among Gay and Bisexual Men: A Preliminary Study in Spain

**DOI:** 10.3390/healthcare11182496

**Published:** 2023-09-08

**Authors:** Xavier Calvet, Leonor M. Cantera

**Affiliations:** Social Psychology Area, Department of Social Psychology, Universitat Autònoma de Barcelona (UAB), 08193 Bellaterra, Barcelona, Spain; leonor.cantera@uab.cat

**Keywords:** sexual violence, intimate partner violence, gay, bisexual, transsexual, victimization, Spain

## Abstract

Sexual violence is an understudied issue in the population of gay and bisexual men, although the existing articles to date demonstrate that it is a problem that merits public attention. This study aims to approach the problem of invisibility around the matter, as well as presenting a number of variables that have been usually overlooked in Spanish research or have not been assessed at all. Lifetime sexual victimization, sociodemographic characteristics, situational characteristics and social support were examined among 550 gay and bisexual males living in Spain using a self-administrated questionnaire. Results analysis show that 90.00% (87.18–92.38%) of participants reported at least one experience of unwanted insinuation, 87.27% (84.19–89.94%) reported at least one experience of sexual coercion, 64.00% (59.83–68.02%) reported at least one experience of sexual assault, and specifically 19.82% (16.57–23.40%) reported being raped during their lifetime. Significant differences have been found between some categories regarding gender identity, sexual orientation, age, race/ethnicity and educational level. Overall, these results showcase sexual violence as a pervasive problem in the Spanish gay and bisexual community.

## 1. Introduction

Sexual violence (SV) is defined by the World Health Organization (WHO) as any unwanted sexual act directed against a person, regardless of the aggressor’s relationship with the victim and the context in which it occurs [1]. The forms of SV may include sexual assault, unwanted sexual advances, sexual coercion or sexual abuse. Thus, the term encompasses a wide variety of circumstances and configurations. For example, it includes rape within marriage and by strangers, as well as abuse of minors and of persons who are mentally or physically incapacitated. Unwanted sexual advances or sexual harassment, refusal to use protection against pregnancy or sexually transmitted diseases (STDs), as well as many others, are also included in their definition of SV. The main reasons for the study of SV are the physical and/or psychological consequences suffered by the victims. Other than the possible injuries produced by any violent experience, unwanted pregnancy and the transmission of STDs, the appearance of phobias, depressive and anxious symptomatology, impoverished self-esteem, sexual dysfunctions, social difficulties and post-traumatic stress disorder are the most prominent consequences for victims of SV [2]. For all these reasons, SV is considered a social emergency that can have serious health consequences.

Historically, SV has been conceptualized as a type of violence carried out by a man against a woman, based on a conception of power that, to a large extent, characterized the second feminist wave. This perspective argues that men use violence against women as an extension of the patriarchal structure and as an expression of control and power over them [3], focusing on the intersection of class and gender systems of oppression [4]. This theory received several criticisms. One of the most notable was from black feminism, which added the intersection of the racial/ethnic axis to take into account the particularities experienced by black women in the patriarchal and capitalist system since the traditional conception neglected such experiences [5]. Gradually, axes of oppression were added, whose intersection explained the particular experiences in terms of race/ethnicity, class, gender, age, sexuality, functional diversity, religion, etc. Specifically, intersectionality, as a branch of feminist theory, is concerned with identifying how various systems of power, operating at multiple levels (e.g., structural and interpersonal), combine and interact to confer disproportionate risk on populations with marginalized social identities [6]. Although stated in a very brief and simplified way, as it is not the main focus of this paper, we can see how the interest in the intersectional study of SV, as opposed to the traditional feminist theory, allows us to explain why some men might suffer victimization experiences. Even so, only a small part of the existing research on intimate partner violence and SV to date is grounded in the theoretical framework and methods of intersectionality [7], so the collective of gay and bisexual men (GBM), the subject of this study, has been particularly neglected. In relation to this, most studies based on the feminist framework have neglected men rape, and this maintains and reinforces patriarchal power relations and hegemonic masculinities [8]. In terms of sexual orientation, we could conclude that the study of SV still has a strong heterosexist bias [9].

Additional resistances can be identified that block the study of SV in this group. First, there exist a number of male rape myths and beliefs, which are organized into three main themes: denial myths (“men cannot be raped”), blaming myths (“men are responsible for being raped”) and trauma myths (“men who are raped do not really suffer discomfort”) [10]. The internalization of these beliefs would result in a decrease in interest in the subject by the scientific community and, consequently, a reduction in the number of studies addressing it. Second, the reluctance on the part of researchers to address unwanted or non-consensual sex in GBM, as well as in any other LGTBIQA+ group, could be related to an attempt to avoid reproducing heterosexist ideas that define sexual minorities as deviant, criminal, predatory, pedophilic, and rapist [11]. In fact, there is an intention on the part of the scientific community to reconceptualize same-gender sexual relationships as healthy, legitimate and non-essentially violent in order to debunk such myths [12], which although necessary, inevitably obscures the violence that might occur within the collective. Third, given the HIV epidemic, there was a great interest and need to focus the study of men who have sex with men on the prevention of HIV transmission. Although this branch of study is interested in exploring SV [13], it tends to prioritize addressing the effects of HIV; an incomplete part of the comprehensive study of SV in the collective. In fact, there is an overwhelming frequency in which certain variables appear in the quantitative literature related to the GBM collective, especially putting the focus on HIV and STDs [14]. The study of these variables is indispensable, but it is also necessary to expand the study of health and well-being among the GBM population to understand many other factors that can be crucial, such as SV.

Although interest in the quantitative and qualitative study of sexual violence in GBM has increased in recent decades, it has been mostly in the context of intragender violence or domestic violence in same-sex relationships [15,16], in addition to prison-associated SV [17] and child abuse [18]. In fact, to our knowledge, there is no Spanish study at present that addresses sexual violence in the GBM collective, although there are a few that do discuss intragender violence [19,20].

There are, however, more international studies on sexual violence focused on the GBM population, regardless of the bond between victim and perpetrator. Some of their results report a 35.2% prevalence of non-consensual sex among men who have sex with men [21], 54.0% prevalence of sexual assault [22], 15.5% of physical SV, sexual assault, and stalking in the past 6 months [23], frequencies for sexual assault of 10.1% for cis gay men and 9.5% for cis bisexual men in the past year [24], 39% lifetime contact sexual violence and 29.2% lifetime non-contact sexual violence for bisexual men and 17.7% and 33.3% for gay men [25], prevalence of SV reaching 31.64% in gay men and 29.44% in bisexual men [26]. Although all the studies mentioned above conclude that SV in the GBM population is drastically higher than in heterosexual cissexual men and similar to heterosexual cissexual women, and also that trans people have a higher risk of victimization, the figures differ greatly among them due to a lack of clear delimitation in the concept of sexual violence (e.g., sexual coercion, sexual aggression, unwanted relationships, intimate partner violence, etc.). It has been noted that while some studies define it very narrowly and only include some of its most obvious manifestations, others include a much wider range of situations [27].

There are a number of factors that indicate that the figures on SV may underrepresent the reality of this group. These factors are related to several barriers that affect GBM in relation to reporting experiences of SV. First, individuals who identify as male are particularly reluctant to admit psychological consequences that are inconsistent with male gender role expectations [28], resulting in poor communication of experiences. Such communication would actually be selective so they would need a higher level of distress to communicate such experiences [27]. In this sense, we should highlight the role of male socialization in understanding the differences in consequences and coping strategies between men and women when victimized [29]. Second, GBM are so stigmatized by their sexual orientation that they may encounter additional difficulties in communicating their experiences of SV for fear of further discrediting themselves [30]. It is also important to take into account the role played by how sexual relations and consent are conceptualized in this group since it is necessary for a SV situation to be labeled as such in order to communicate it in a relevant way. In fact, the most common method used by GBM to give consent is not to resist [31]. This makes it difficult to differentiate a consensual situation from a violent one, both for the perpetrator and the victim. There are some perspectives that claim that there is a “culture of silence” in the collective [32] that accepts, normalizes, and consequently obscures SV. Although it seems that some people in the GBM collective accept all these situations and do not cause them discomfort, some others are resisting such normalization [11], and they are having trouble labeling some experiences as sexual violence because they have them internalized. There are a number of factors that would result in an intentional ambiguity in defining what is and what is not SV in sexual minorities. Such factors would include stereotypes like gay men always want to have sex, gender norms regarding masculinity, a particular vision of queerness that could, in fact, foreclose the role of power dynamics, and the risk of social isolation when someone refuses sexualized social relations [33].

The aim of this study is to make SV visible for GBM individuals in Spain. To do so, the prevalence of victimization and the frequency in which certain specific violent situations occur, as well as some of their main characteristics have been studied. Emphasis has been placed on the communication of experiences; it is of interest to know to what extent victims communicate and to whom they turn in order to understand the low visibility of the problem and to propose preventive strategies and interventions. It is not the purpose of this study to pathologize the sexuality of men who have sex with men nor to suggest that violent behaviors are the primary framework from which we can generalize all GBM sexual experiences. Instead, it is intended to give voice to the discomfort experienced by some GBM individuals in order to promote a discourse that supports positive sexual experiences and the emotional well-being of GBM individuals.

## 2. Materials and Methods

### 2.1. Study Design and Data Collection

This is a cross-sectional study aimed at obtaining quantitative information through a self-administered questionnaire. Participation was anonymous and voluntary and aimed at the GBM population living in Spain. The final sample consisted of 550 respondents.

### 2.2. Measurement Tool

The questionnaire was built using Google Forms v0.8 software. The definition of SV from the WHO was used, through which SV situations were extracted and transformed into items [1]. Thus, the pilot version of the questionnaire consisted of these items.

It was administered for the first time in a small sample of GBM who had suffered some type of violence (10 participants) to ensure that there were no problems with the comprehension of the items and that the participants could give their opinions in general. Interviews were conducted with questions about the appropriateness of the items for the GBM collective in order to make them more sensitive to the reality of this particular group. Based on their recommendations, the questionnaire was reconstructed in such a way that new situations of sexual violence and more response possibilities were added to define their characteristics, as well as eliminating those questions that were labeled as uncomfortable or out of place. It was also decided to divide the experiences into two separate blocks (experiences with and without physical contact) to ensure that the mildest situations were asked first and that the most sensitive ones were presented little by little.

The final version of the instrument was intended not only to collect data but also to serve as a tool for participants to share their perceptions of the experience at length if they so wished. It can be seen as an attempt to provide a tool with which to express themselves, given the invisibility and problems inherent in communicating the experiences of SV in this group. The internal consistency of the questionnaire was α = 0.87 for the total scale.

### 2.3. Variables

#### 2.3.1. Socio-Demographic Information

Information on gender identity, sexual orientation, race/ethnicity, level of education to date and autonomous community of usual residence was collected. Questions on gender identity and sexual orientation were part of the exclusion criteria and sent some participants to the end of the questionnaire according to their answers. The responses referring to the autonomous community of habitual residence covered all those autonomous communities and cities that are part of the Spanish state.

To avoid bias, the questions that formed part of the exclusion criteria were asked before explaining the goals of the study and the target population. In this way, we tried to reduce the probability that participants who did not meet the admission criteria could falsify their identity, since once these variables were answered, participants were directed to the end of the questionnaire and could not participate again from the same email address.

#### 2.3.2. Prevalence and Frequency of SV Experiences

To study the prevalence and frequency of perceived SV, participants were asked to what extent they have experienced a series of violent situations. The information about the frequency offers us not only information about what proportion of the sample has suffered each situation, but also informs us about the extent to which each participant has suffered those experiences. These frequencies were recorded on a Likert scale (1 = Never/2 = Once/3 = Sometimes/4 = Often/5 = Many times). Contrary to other studies, we avoided asking about the exact number of times each situation occurred. Firstly, it is difficult for participants to get the count of each situation right, especially when dealing with potentially traumatic experiences or almost normalized experiences. Secondly, because this study focuses on the victim’s perception: a situation may have happened five times but for one person it may be labeled as “sometimes” and for another as “many times”.

The blocks of experiences have been differentiated as done by other studies to allow comparison into unwanted insinuation, sexual coercion, and sexual assault. The situations that were part of unwanted insinuation were (1) Unwelcome sexually explicit comments (Comments) and (2) Inappropriate exhibition of genitalia (Exhibition). Sexual coercion included (3) Excessive insistence to have sex (Insistence), (4) Verbal recrimination (e.g., insults) motivated by the refusal to have sex with the perpetrator (Verbal Recrimination), (5) Blackmail and threats in order to convince the victim to have sex (Blackmail and Threats), (6) Physical punishment motivated by the refusal to have sex with the perpetrator (Physical Punishment), (7) Victim incapacitated (physically or mentally) to give consent or to resist (Self Incapacitation) and (8) Victim intentionally drugged by the perpetrator for sexual purposes (Chemical Submission). Sexual assault included (9) Unwanted touching with sexual intent (Touching), (10) Attempted rape: victim managed to escape the situation (Attempted rape), (11) Forced physically without penetration, whether oral, anal or vaginal (Forced without penetration) and (12) Forced physically with penetration, whether oral, anal or vaginal (Rape). Items (13) Intentional and unwanted injuries during sex (Injuries), (14) Image-based sexual violence: the perpetrator shows or posts sexual images of the victim without consent (Images) and (15) Refusal to use protection or not using it without the victim’s consent (Protection) did not correspond to any of the previous blocks.

To avoid ambiguity to the greatest extent possible, the items have been written specifying that there must be a feeling of discomfort. In this way, an attempt has been made not to record those situations that were appropriate for the context. For example: “… without my consent”, “… and I did not want it”, “… and I think it was inappropriate”.

If any participant answered “Never” in all situations, they were redirected to the end of the questionnaire because the rest of the questions were not applicable.

#### 2.3.3. Situational Characteristics

Some information about the violent person has been examined. In order to keep the questionnaire brief, these characteristics have been recorded, as well as all of the following, for the overall situations of SV instead of asking for the characteristics of each of the situations separately. For this reason, it was reminded in these sections that in cases of having experienced several situations of violence or being protagonized by different perpetrators, the participant should mark all the relevant answers.

Regarding the age difference with the perpetrator, exact numbers were again avoided with the same justification: victims may have problems remembering all the exact ages (especially in cases where there are several abuses, several abusers or if he/she was incapacitated) and an age difference of five years, for example, may or may not be a clear age difference depending on the person and their relationship with the perpetrator. The gender identity of the perpetrator and the bond with the victim at the time of the abuse were also explored.

It was asked if any of the violent situations had been group aggressions and if any of them had been carried out by the same person more than once. Information about the place where the violent situation occurred was also collected.

#### 2.3.4. Communication and Social Support

Participants were asked if they have shared their experiences, and if so, to whom they have shared them. A general question was asked about the potential usefulness of support groups (if they existed) for GBM individuals who have suffered violence. It was evaluated on a Likert scale (1 = Not useful at all/5 = Very useful).

### 2.4. Participants

The participants who were part of the sample met the following inclusion criteria: (1) identify as male (regardless of being cissexual or transsexual), (2) identify as homosexual or bisexual, and (3) live in an autonomous community or city in Spain on a regular basis. Any other answer to the questions related to gender identity, place of habitual residence or sexual orientation were discarded. Thus, if a total of 1220 responses were recorded, the initial sample included responses from the 690 participants who met the inclusion criteria. To ensure representation of the territorial distribution of the population, the number of participants from each autonomous community that was included in the study was proportional to the population of men in each community in relation to the Spanish total based on data from the National Institute of Statistics [34]. Thus, some responses from those autonomous communities and cities that were overrepresented were randomly discarded. The final sample that would later be analyzed was composed of the responses of 550 participants.

These participants were obtained from the dissemination of the questionnaire through social networks (Instagram and Twitter) with the help of various Spanish LGTBIQA+ associations and through a profile on Grindr, a dating application for GBM, which contained the link to the questionnaire.

## 3. Statistical Analysis

The statistical program Stata version 14.2 was used for data analysis. A univariate analysis has been conducted for the experiences of SV, situational characteristics and communication and social support. Bivariate analysis is meant to uncover the differences between sociodemographic groups regarding SV prevalence and frequency.

The responses to each situation of SV were interpreted dichotomously (1 = No; 2, 3, 4, 5 = Yes) for the prevalence study, following the criterion of “tolerance 0”. That is, any violent situation was considered to be present if the participant responded with any value other than 1. Confidence of 95% was used for the elaboration of confidence intervals (CI) for the univariate analysis. Analysis of variance (ANOVA) was carried out to determine the differences between groups regarding experiences of SV according to their sociodemographic characteristics: gender identity, sexual orientation, age, race/ethnicity and educational level for the bivariate analysis. Statistical significance was interpreted with *p*-value < 0.05 from Student’s t calculation. Bonferroni adjustment was calculated separately for the analysis of each group. It should be noted that some category values do not have a large enough sample size to generalize the results. Situational characteristics and social support were examined from a descriptive approach.

## 4. Ethics Considerations

The questionnaire construction and management considered the ethical aspects of the Code of Good Practice of the Autonomous University of Barcelona and the recommendations of the European Charter for Researchers, based on the fundamental principles of anonymity, freedom, honesty and responsibility. The participants were aware of the objectives of the study and gave their consent to participate in it.

## 5. Results

### 5.1. Sociodemographic Characteristics

Table 1 shows the sociodemographic characteristics of the participants who were part of the final sample. Most of them were cis homosexual white young men with a high educational level.

### 5.2. Prevalence and Frequency of SV Experiences

Table 2 provides a summary of the prevalence and frequency of occurrence of each of the vs. items and blocks presented in this study. The most prevalent situations for the GBM group in Spain are unwanted insinuations (87.18–92.38%), especially consisting of explicit sexual comments that are inappropriate (83.80–89.61%). The figures on the denial of the use of protection (42.67–51.18%) are also relevant considering the great number of STDs in this collective. The prevalence of sexual assault (59.83–68.02%) is especially high, with a 54.30–62.70% prevalence of situations of unwanted touching. Figures on rape prevalence are also alarming, ranging from 16.57% to 23.40%. Results on frequency are coherent with those found in the prevalence analysis.

### 5.3. Situational Characteristics

The results on the situation and characteristics of the perpetrator are presented below (Table 3). It is of interest to note that the perpetrator is a stranger to the victim in most cases (63.65%) and that their most usual characteristics are being a cis man (95.09%), older than the victim (68.96%) and online using gay dating apps (58.15%); 9.63% of the sample informs us to have experienced group violence and 29.86% to have experienced violent situations with the same perpetrator more than once. Most participants were victimized when they were 19–24 years old (58.94%).

### 5.4. Communication and Social Support

Finally, the responses regarding communication and social support are reflected in Table 4. The usefulness of the support groups for the participants was rated positively with a mean of 4.26 (95% CI: 4.17–4.36). Of all the participants of the study who have suffered any kind of violence, most of them did not share all their experiences (67.58%). Most of them trusted their friends to do so (84.48%).

### 5.5. Experiences of SV × Sociodemographic Characteristics

The bivariate analysis that has been conducted regarding SV experiences vs. Sociodemographic characteristics is shown below with the symbols * = *p*-value < 0.05, ** = *p*-value < 0.01, *** = *p*-value < 0.001.

In terms of prevalence, trans people report a higher prevalence than cis people in the situations of *Incapacitated* ** and *Rape* **, and in *Sexual Assault* *. Bisexuals also reported a higher prevalence than homosexuals in situations *Forced without penetration* *. The age variable had the greatest effect on the items referring to situations of SV, with the youngest respondents showing a significantly higher prevalence. This is the case for the items *Comments* ***, *Exhibition* *, *Insistence* ***, *Blackmail and threats* ***, *Verbal recrimination* ***, *Rape* *** and in the groupings *Unwanted insinuation* *** and *Sexual coercion* **. In addition, people with no studies or only compulsory secondary education have reported significant differences showing a higher prevalence of SV, in most cases compared to all other categories. This is the case for the items *Chemical Submission* *, *Injury* *, *Physical Punishment* *** and *Forced without penetration**. Finally, with respect to the Race/Ethnicity categories, differences were only found in the comparison between Latinos and whites, with Latinos reporting a higher prevalence of SV. Such differences are found in the variables *Chemical Submission* * and *Attempted Rape* *.

Regarding the frequency of occurrence, the results are similar. Trans people reported higher frequencies in the item *Rape* *. Bisexual people in the items *Forced without penetration* ***. Younger participants in the items *Comments* ***, *Exhibition* ***, *Insistence* ***, *Blackmail and Threats* *, *Verbal Recrimination* **, *Rape* *, the groupings *Unwanted Insinuation* *** and *Sexual Coercion* **. People with no education or with compulsory secondary education in the items *Touching* *, *Injuries* *** and *Forced without penetration* **, the groupings *Sexual coercion* ** and *Sexual assault* **. Finally, Latinos reported a higher frequency of *Chemical Submission* *.

## 6. Discussion and limitations

First, it is necessary to review the characteristics of the sample to see how representative the results really are. Although it was intended to be representative of the Spanish territory, it is not representative of most sociodemographic issues, so some categories are underrepresented: the vast majority of participants were cissexual, homosexual, with a higher level of education, white and young. In this sense, the sample of trans men is particularly small, so the results should be interpreted with caution, although many articles agree that this group receives more violence than cis people [35]. The sample of people with special education and compulsory secondary education/no education is also insufficient, as well as those of all race/ethnicity categories except for white and Latino men. All of these difficulties were to be expected, since first of all, very large samples are required to do intersectional studies [36] and also, the GBM population is very much in the minority, so it is difficult to get participants, and consequently, to make it large and representative. Even so, the considerable sample size of this study is considered a strength for the reliability of the results for the GBM population in Spain.

Although one of the objectives of the questionnaire was to ensure its brevity, it would be much closer to reality to include non-exclusively categorical measures of sexual orientation and gender. Taking into account that from a constructivist perspective gender is constructed in each social interaction to the extent that heterosexist social conventions are repeated (or not) [37], it would be interesting to understand to what extent the tendency taken by this performance affects the directionality of the perpetration-victimization, rather than focusing exclusively on gender identity. Indeed, in that sense, the terms gay and lesbian, beyond signifying only preferences in the object of desire, could be well-recognizable social identities, perhaps authentic and proper genders [38]. There is a need to advance the study of gender from a non-categorical queer perspective in order to understand the phenomena studied in all scientific disciplines in a comprehensive way; specifically, in the field of SV. Exploring beyond a binary definition of gender could help to make violence against non-heterosexual individuals more visible.

Given the variability and multiplicity of definitions and interpretations of SV, we cannot assure construct validity in this study, although the items that form part of the questionnaire were elaborated on the basis of what the WHO defines as SV [1] and were modified under the evaluation of GBM individuals. The figures on the prevalence of SV in the group are extremely high, which is not surprising considering that a wide range of situations of greater or lesser severity have been examined and that this is a study on lifetime victimization. However, the numbers could have been magnified due to the method of data collection, as most participants have been recruited by a GBM dating app and may be more likely to have experienced SV situations than those who do not use them. Comparison with other studies is not possible, firstly because the instrument for exploring SV situations is different, but also because some studies have only focused on childhood or adulthood experiences, while others have only recorded situations that occurred during a period of time (usually during the 6 months or a year before participating in the study). In addition to all these differences, it is worth noting the different conceptions of the concept of SV or sexual assault [27]. The only studies with which comparisons could be made report a 54.0% [22] and 35.2% [39] lifetime prevalence of sexual assault, although the questionnaire they use is very different, the definition of sexual aggression is slightly different, and they are from different countries. Those figures are lower than the prevalence found in this study, which is around 64% for sexual assault. On the other hand, the study on frequency is quite atypical in a work of these characteristics, so there is no quantitative data with which to compare the results. However, it is consistent with the violent behaviors that have been labeled previously as normalized and that manifest themselves as unwanted sexual advances (comments, exhibition, insistence) and touching, which are precisely the variables that have shown the highest frequency averages (>2), as well as verbal recrimination [11].

Results show that bisexual men, transgender men, those with a low educational level, and Latinos are more at risk of experiencing SV. This fact could be related to the concept of homonormativity, which refers to the privileges that some LGTBIQA+ people have for conforming to heterosexual norms, so that those cis homosexual men, without disability, with normative bodies, gender expressions and affective relationships would get more social recognition, so those “deviant” people would be more vulnerable to suffer violence [40,41]. Still, the sociodemographic characteristic that most significantly affects victimization is age. In fact, it is curious that, as this is a lifetime victimization study, it is the youngest people who report the most violence in a very significant way. Although it is not possible from this study to establish causal relationships, it could be due to (1) generational differences in the prevalence of SV, probably due to the emergence of the internet and social networks or the existence of more nightlife spaces for GBM, (2) a greater ability to label violent situations correctly due to the influence of the fourth feminist wave [42] in younger people, with movements such as #MeToo or (3) a memory bias that would make it difficult for older people to remember violence in their youth, especially for mild experiences of SV.

Although the variable of victimization age, to our knowledge, has not been studied in SV within the collective, it has been studied in studies of intragender violence. The age variable is found to be the strongest and most consistent characteristic in relation to the victimization of any type of violence among gay male partners, a fact that they attribute to the acquisition of internal and external resources with age that are protective against violent situations [43]. However, in our results, it is much more frequent that the perpetrator is older than the victim. Considering that the majority of the sample in this study were young people, more studies are needed in the field of SV on GBM to determine whether age could be a power resource for the perpetration of violence against younger people.

In relation to spaces, the prevalence of SV in dating apps is particularly high. In reference to this, we should consider the manifestations of rape culture in such spaces, particularly through Grindr, the geolocation-based social network most used by men who have sex with men. It would manifest itself through violent comments and unsolicited nude photos, and extend beyond online interactions in the form of sexual assault, sexual coercion, and image-based sexual violence [44]. This is consistent with articles that emphasize how Grindr encourages its users to see each other as objects to be consumed and discarded at will [45], a fact that could be considered an ethical danger in that objectifying and instrumentalizing others for one’s own sexual pleasure could result in avoidance and closure towards otherness [46]. The effects that this whole scenario may have on the reproduction of vs. is evident, as it produces sexual encounters in which pleasure is centered on oneself and not on the interaction between the participants, a fact that could hinder communication and the interpretation of consent signals. The data from this study invites us to rethink the design, structure and format of apps for encounters between men who have sex with men, inasmuch as it is one of the spaces in which most situations of SV occur. Also noteworthy is the prevalence of SV in nightclubs and gay bars, which could be explained by the effect that alcohol and other drugs have on the perpetration of violence by predisposed individuals [47], as well as on the vulnerability to victimization [48]. A study of the expectations of the people who frequent these spaces could be of interest in order to investigate the relationship they may have with the perpetration of violent situations.

Although several participants added in the “Other” option spaces such as saunas, chills, cruising areas, dark rooms, etc., they have not been addressed in this study because it would be too hasty to draw conclusions about SV from a quantitative point of view and could run the risk of stigmatizing these sexual practices. These are situations that could be understood as violent in and of themselves, which is why it is necessary to study them in-depth, from a qualitative framework, in order to investigate how consent operates in these contexts [49].

Almost all perpetrators, as reported by the victims, are cis males older than the victim. The bond with the perpetrator is one of the most interesting variables of the study, as it shows that the vast majority of violence is perpetrated by strangers. This implies that it does not make sense to study SV on GBM only in the area of intragender violence or intimate partner violence in homosexual relationships. The figures are very similar to those found in another study [21], especially high for strangers (33.3% vs. 63.65% in this study), casual partners/dates (29.40% vs. 33.99% in this study) and acquaintances (15.70% vs. 28.68% in this study). For future studies, it would be interesting to include more types of bonds, since the LGTBIQA+ group is very rich in forms of relationships (polyamorous relationships, exclusively sexual relationships, open relationships, etc.) and in this way, a representation closer to reality would be achieved.

Finally, the fact that the vast majority of victims communicate their experiences to friends rather than to other links refers to the phenomenon of *peers’ communication* and is consistent with existing literature [16]. In that sense, victim support groups could be a useful tool to promote such communication.

It is important to note that all answers regarding situational characteristics and social support have been registered for all the situations of SV that the participants may have lived at once, so this study is not capable of attributing this data to any specific situation of SV. For example, we cannot tell from these figures if most rapes in particular are carried out by strangers or by acquaintances. Those difficulties were accepted to ensure the questionnaire’s brevity, and therefore, to achieve the largest possible sample. As this is a preliminary study, it shows an overview of the problem and encourages other studies to further investigate the specific topics discussed above.

Although it is inevitable that the study presents a series of limitations, it is important to emphasize the importance of being a pioneer in the study of SV on GBM living in Spain, as well as its usefulness as a tool for communication and expression for the victims. It is necessary to promote the study of sexual violence in this collective, as it is for all the letters of the LGTBIQA+ group. Addressing this issue, and consequently constructing prevention and intervention strategies focused on the particularities of this collective, would have important positive health outcomes. Not only because it would improve physical well-being by preventing STDs and the physical harm a violent situation may involve, but because it would also have a positive impact on psychological well-being.

## 7. Conclusions

This paper reveals that sexual violence is common among gay and bisexual men. Findings show that prevalence figures depend on sociodemographic characteristics like gender identity, sexual orientation, race/ethnicity and educational level. Prevalence would also be higher in certain situations and spaces. Some of the perpetrators’ and victims’ most usual characteristics have also been examined. Furthermore, it has come to light that those who have been affected face challenges when it comes to sharing their experiences with others and that they mostly rely on their peers to do so.

Epidemiological research and interventions should take into account the intersections between gender identity and sexual orientation to better tailor prevention and treatment in this collective. Given the invisibility and stigma associated with this issue, this study highlights the usefulness that support groups could have in facilitating victims’ communication. As a preliminary study, this paper could be useful for further in-depth research on the topics discussed above, such as sociodemographic factors related to the risk of victimization or the role of external situational characteristics to specific situations of sexual violence. It is necessary to promote the study of sexual violence in this group to understand the power dynamics that could underlie these situations.

## Figures and Tables

**Table 1 healthcare-11-02496-t001:** Sociodemographic data of the participants.

Variables	n (%)
**Sexual orientation**	
Homosexual	413 (75.09%)
Bisexual	137 (24.91%)
**Gender identity**	
Cis man	538 (97.82%)
Trans man	12 (2.18%)
**Educational level**	
Higher education (University, CFGS)	432 (78.55%)
Post-compulsory secondary education (Bachillerato, CFGM)	81 (14.73%)
Secondary education (ESO)/No studies	24 (4.36%)
Special regime education	13 (2.36%)
**Race/Ethnicity**	
White	473 (86.00%)
Latino	65 (11.82%)
Bi- or multiracial	8 (1.45%)
Black	1 (0.18%)
Middle East	1 (0.18%)
**Age**	
25–34	221 (40.18%)
18–24	191 (34.73%)
35–44	90 (16.36%)
45+	48 (8.73%)
**Autonomous Community/City**	
Andalucía	99 (18.03%)
Cataluña	90 (16.36%)
Comunidad de Madrid	77 (14.00%)
Comunidad Valenciana	59 (10.73%)
Galicia	31 (5.64%)
Castilla y León	28 (5.09%)
Islas Canarias	25 (4.55%)
País Vasco	25 (4.55%)
Castilla–La Mancha	24 (4.36%)
Región de Murcia	18 (3.27%)
Aragón	16 (2.91%)
Islas Baleares	14 (2.55%)
Extremadura	12 (2.18%)
Principado de Asturias	11 (2.00%)
Navarra	8 (1.45%)
Cantabria	7 (1.27%)
La Rioja	4 (0.73%)
Ceuta	1 (0.18%)
Melilla	1 (0.18%)

Note. The percentages on autonomous communities/cities differ a maximum of ±0.09% from the INE figures about male population.

**Table 2 healthcare-11-02496-t002:** Prevalence and frequency for each situation of SV.

VS Variable	Total Prevalence	[95% CI] Prevalence	Average Frequency	[95% CI] Frequency
**Unwanted insinuation**	90.00%	87.18–92.38%	3.01	2.91–3.10
Comments	86.91%	83.80–89.61%	3.07	2.98–3.16
Exhibition	76.91%	73.16–80.37%	2.95	2.83–3.06
**Sexual coercion**	87.27%	84.19–89.94%	1.78	1.72–1.83
Insistence	82.91%	79.50–85.96%	2.98	2.89–3.08
Verbal recrimination	58.91%	54.67–63.05%	2.17	2.07–2.27
Blackmail and threads	34.36%	30.39–38.50%	1.63	1.54–1.71
Self-incapacitation	31.09%	27.24–35.14%	1.54	1.46–1.61
Chemical submission	12.91%	10.22–16.00%	1.18	1.14–1.23
Physical punishment	10.00%	7.62–12.82%	1.16	1.11–1.20
**Sexual assault**	64.00%	59.83–68.02%	1.55	1.50–1.60
Touching	58.55%	54.30–62.70%	2.15	2.06–2.24
Attempted rape	27.27%	23.59–31.20%	1.38	1.32–1.44
Forced without penetration	23.81%	20.31–27.60%	1.36	1.30–1.42
Rape	19.82%	16.57–23.40%	1.30	1.24–1.36
**Injuries**	25.27%	21.69–29.12%	1.36	1.30–1.42
**Images**	25.64%	22.04–29.50%	1.47	1.39–1.55
**Protection**	46.91%	42.67–51.18%	1.86	1.77–1.95

Note. The blocks Unwanted insinuation, sexual coercion and sexual aggression have been calculated from the average of the items that comprise them.

**Table 3 healthcare-11-02496-t003:** Situational characteristics of the SV experiences.

Variables	n (%)
**Bond with the perpetrator**	
Stranger	324 (63.65%)
Casual partner/date	173 (33.99%)
Acquaintance	146 (28.68%)
Friend	65 (12.77%)
Long-term relationship	41 (8.06%)
Ex-casual partner/date	25 (4.91%)
Family member	21 (4.13%)
Ex-long-term relationship	12 (2.36%)
Workmate	8 (1.57%)
Other	5 (0.98%)
**Perpetrator age**	
Older	351 (68.96%)
Similar ages	216 (42.44%)
Younger	33 (6.48%)
Don’t know	20 (3.93%)
**Perpetrator gender identity**	
Cis man	484 (95.09%)
Don’t know	21 (4.13%)
Cis woman	12 (2.36%)
Trans woman	8 (1.57%)
Trans man	7 (1.38%)
Non-binary	4 (0.79%)
**Group violence**	
No	460 (90.37%)
Yes	49 (9.63%)
**Recidivist violence**	
No	357 (70.14%)
Yes	152 (29.86%)
**Space/situation**	
Online (dating applications)	296 (58.15%)
Perpetrators home	203 (39.88%)
Gay disco/bar	195 (38.31%)
Victims home	130 (25.54%)
In the street	107 (21.02%)
Online (other applications)	102 (20.04%)
Date	98 (19.25%)
Non-gay disco/bar	89 (17.49%)
Someone else’s house	70 (13.75%)
Education center	24 (4.72%)
Other	12 (2.36%)
**Victimization age**	
19–24 years old	300 (58.94%)
16–18 years old	176 (34.58%)
25–34 years old	164 (32.22%)
Under 16 years old	83 (16.31%)
35–44 years old	45 (8.84%)
45+ years old	12 (2.36%)

Note. The sum of the percentages for each variable is in some cases >100% because participants could choose more than one response option.

**Table 4 healthcare-11-02496-t004:** Communication and social support of the participants.

Variables	n (%)
**Communication**	
Yes, but not all situations	344 (67.58%)
Yes, all of them	109 (21.41%)
No, none of them	56 (11.00%)
**Social support**	
Friends	430 (84.48%)
Long-term relationship	120 (23.58%)
Mental health professional	77 (15.13%)
Brother/sister	46 (9.04%)
Mother	33 (6.48%)
Other family member	26 (5.11%)
Father	13 (2.55%)
Support group	13 (2.55%)
Teacher or similar	4 (0.79%)
Other	4 (0.79%)
Ex-long-term relationship	1 (0.20%)

Note. The sum of the percentages for each variable is in some cases >100% because participants could choose more than one response option.

## Data Availability

The data sets used and analyzed in the current study are available from the corresponding author on reasonable request.

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
