# Peer review of "Prevalence and Characteristics of Sexual Victimization among Gay and Bisexual Men: A Preliminary Study in Spain"

_healthcare, 2023, doi:10.3390/healthcare11182496_

Round 1
Reviewer 1 Report
The authors present a good research paper with a topic of crucial importance to be able to visualize the magnitude of a barely explored problem such as sexual violence in the population of gay and bisexual men. Specifically, the strength of this research lies in, as commented by the authors of this manuscript, the importance of being a pioneer in the study of SV on GBM living in Spain, as well as its usefulness as a tool for communication and expression for the victims of this sexual violence.
The objective of the manuscript is to make SV visible for GBM individuals in Spain. Despite being a good research paper, well carried out and argued, some issues, which I detail more specifically below, could improve the quality of this manuscript for its academic publication

Author Response
We really appreciate your review for our paper. All changes regarding your review have been highlited in yellow in our manuscript. Please see the attachment.
Best regards.

Reviewer 2 Report
1. Near the bottom on page 7, replace “Student T…” with “Student t…”
2. On page 11, line#375-#376, authors will mention the prevalence in this study for comparison.
3. On page 11, line#383, authors will mention the frequencies in this study and other studies for comparison.
4. On page 11, line#442-#443, authors will report the figures in this study and others for comparison.
5. In Conclusions (page 12), authors will suggest some prevention and treatment based on their findings in this study.
Author Response
We really appreciate your review for our paper. We considered all your comments, you can read our obsevations below. All changes regarding your review have been highlited in green in our manuscript.
- Done.
- Done.
- There is no possible comparison to be made, because the other study uses a qualitative approach. However, we added a clarification explaining that a situation of SV was considered specially frequent in our study if the frequency mean was >2. This would mean that on average, all participants have lived this specific situation more than once.
- Done. However, figures are slightly different because the other study has a sample of 51 participants, which is very small compared to our 509 participants. What we were trying to highlight was that both studies share the same "ranking".
- We alredy suggest support groups as a possible treatment option based in our findings. As it is a preliminary study that includes a lot of variables and hypotheses, we don't have enough specific data to firmly support a specific prevention or treatment. The aim of this study is to make sexual violence visible for the gay and bisexual men collective and, therefore, to promote more specific reaserch regarding the topics that are included in the manuscript.
Best regards.
Reviewer 3 Report
Prevalence and characteristics of sexual victimization among gay and bisexual men: a preliminary study in Spain
This is a very interesting and important contribution to the field. The article is well written and scientifically sound, as well as pioneering in Spain. Still, some changes would be beneficial before it can be published:
1. Instead of “variables”, include all information under the section “measurement instruments”.
2. Dissemination of the link must be clearly informed. How were men reached out?
3. How do gay and bisexual males living in Spain describe the cultural diversity of the country? Is it the same living in Barcelona or in a small town in Galicia?
4. The limitations section must be further described. There are many other limitations that were not mentioned.
5. Implications, especially health and social implications in Spain, sensitive to its cultural characteristics must be further discussed.
Best wishes.
Author Response
We really appreciate your review for our paper. We considered all your comments, you can read our obsevations below. All changes regarding your review have been highlited in cyan in our manuscript.
- A different format was suggested by another reviewer. We tried our best to explain the instrument construction under the section 2.2. Measurement tool and we changed the position of the 2.3. Variables right after. Let us know if you agree with these changes in the lastest version of the manuscript.
- Done.
- This paper does not attempt to look for differences between rural and urban environments. The information we gathered was exclusively focused on the Autonomous Communities and it can't differentiate wether participants are participating from cities or small towns. We believe such research should be carried by a qualitative approach.
- We mixed the discussion and the limitations because we considered that we needed to take into account the shortcomings of our study in order to better interpretate the results. We mentioned limitations regarding the questionnaire construction, the selection of variables, construct validity, the lack of diversity in the sample and the data collection method. We just added a paragraph explaining a limitation we overlooked regarding response format in this lastest version of the manuscript. Let us know if you appraise more limitations to be considered.
- Done. Since it's an issue that affects the whole GBM community, regardless of the country they live in (even though there aren't many published articles discussing the issue), we do not understand how can we approach Spanish cultural characteristics in this topic.
Best regards.
Round 2
Reviewer 1 Report
Title
Prevalence and characteristics of sexual victimization among gay and bisexual men: a preliminary study in Spain
The authors have followed the proposed instructions and the manuscript has improved substantially. However, some aspects are identified that must be improved for its final acceptance:
Line 283
Ethics considerations
If the previous section is 3. Statistical analysis, the next one would be 4. Ethics consideration and not again 3. Ethics considerations
Line 294
Table 1. Sociodemographic data of the participants.
Table 1 is cut and the results cannot be clearly seen.
Line 307
Table 2. Prevalence and frequency for each situation of SV
I do not think it is mandatory to order the figures from the highest to the lowest, the reader understands if a % is less or greater than another regardless of this, what he may not understand is a table in which the main categories do not have their subcategories just below each of them, so I insist on my recommendation to the authors that they write this table like this, after each category or main variable its components or subcategories with their % and corresponding values in the rest of the columns.
Line 330
Experiences of SV x sociodemographic characteristics
This subsection should be 4.4. and not 4.3, since the previous one is 4.3. Communication and social support
Author Response
All changes were applied.
- Line 283: Done.
- Line 294: Done.
- Line 307: Done.
- Line 330: Done.
We appreciate your comments. Kind regards.